# Effects of Cross-Training on Medical Teams’ Teamwork and Collaboration: Use of Simulation

**DOI:** 10.3390/pharmacy7010013

**Published:** 2019-01-19

**Authors:** Ashley R. Hedges, Heather J. Johnson, Lawrence R. Kobulinsky, Jamie L. Estock, David Eibling, Amy L. Seybert

**Affiliations:** 1Department of Pharmacy Services, University of Utah Health, Salt Lake City, UT 84132, USA; 2Department of Pharmacy and Therapeutics, School of Pharmacy, University of Pittsburgh, Pittsburgh, PA 15213, USA; johnsonhj@upmc.edu (H.J.J.); lrk26@pitt.edu (L.R.K.); seyberta@pitt.edu (A.L.S.); 3Department of Pharmacy Services, University of Pittsburgh Medical Center (UPMC), Pittsburgh, PA 15213, USA; 4VA Pittsburgh Healthcare System, Center for Medical Product End-User Testing, Pittsburgh, PA 15213, USA; jamie.estock@va.gov; 5Department of Otolaryngology, School of Medicine, University of Pittsburgh, Pittsburgh, PA 15213, USA; david.eibling@va.gov; 6VA Pittsburgh Healthcare System, U.S. Department of Veterans Affairs, Pittsburgh, PA 15213, USA

**Keywords:** simulation, interprofessional education, teamwork, cross-training

## Abstract

Previous research in the US Navy demonstrated that cross-training enhances teamwork and interpersonal collaboration. Limited data exists on cross-training effectiveness in medical education. This research aimed to assess whether cross-training would have similar effects on medical teams. A multidisciplinary pair of resident participants—consisting of one physician and one pharmacist—was randomly assigned to cross-training or current training condition. The training experience involved one video-based content module (training a pharmacist’s task of pharmacokinetic dosing and a physician’s task of intubation) and one simulation-based practice scenario (collaborative treatment of an unstable critically ill simulated patient). Interprofessional pairs randomized to cross-training condition participated in both the content module and practice scenario in the alternative professional role whereas pairs randomized to current training condition participated in their own professional role. Pairs also participated in pre- and post- training assessment scenarios in their own professional role. Teamwork and interprofessionalism were measured immediately following assessment scenarios. Knowledge assessments were conducted at the start and end of the scenario sequence. Multidisciplinary pairs experiencing cross-training showed a significant improvement in teamwork (increased by 6.11% vs. 3.24%, *p* < 0.05). All participants demonstrated significant improvement in knowledge scores (increase of 14% cross-training, *p* < 0.05, and increase of 13.9% control, *p* < 0.05). Our project suggests that cross-training can improve teamwork in interprofessional medical teams.

## 1. Introduction

Current medical care has evolved to have a strong interprofessional structure, suggesting that teamwork amongst healthcare professionals is vital to assuring patient safety. Simply put, “teams make fewer mistakes than individuals” [1]. Therefore, how to train effective teams has become a topic of interest among healthcare educators [2,3]. In fact, a growing number of national organizations and accreditation councils have highlighted the need to evaluate trainee competencies in the areas of interprofessional teamwork and communication [4,5,6,7]. Both the Accreditation Council of Graduate Medical Education and the American Society of Health-System Pharmacists Residency Accreditation Standards require residents to demonstrate an ability to work effectively within interprofessional teams [4,5,6,7]. Moreover, interprofessionalism, defined as two or more healthcare professions collaborating or coordinating with a shared goal of enhancing patient care, is being taught earlier by being integrated into core standards for medical and pharmacy student curriculums alike [8].

In a recent report from the Agency for Healthcare Research and Quality (AHRQ), the authors recommend that medical team training could be enhanced by leveraging existing research on other instructional methods [1]. The AHRQ specifically highlights the extensive research conducted by Salas and colleagues on team training for the US Navy, in particular, the concept of cross-training [1,9,10]. Cross-training is defined as “a strategy in which each team member is trained on the tasks, duties, and responsibilities of his or her fellow team members. The goal of this type of training is to provide team members with a clear understanding of the entire team function and how one’s particular tasks and responsibilities interrelate with those of the other team members” [10]. Research has shown that cross-training significantly improves overall teamwork, by enhancing interpersonal collaboration and communication [9,10]. Additionally, teams that have been cross-trained have demonstrated better performance in a high-stress environment [10]. The goal of this research was to evaluate the impact of cross-training on teamwork, interprofessionalism and knowledge in multidisciplinary medical teams. We hypothesized that both groups (i.e., cross-trained and control) would show improvement in their teamwork and interprofessionalism scores post-training, but we expected that the cross-trained teams would show a greater improvement in both teamwork and interprofessionalism, as compared to the control group. 

## 2. Materials and Methods 

This randomized study was approved by the institutional review board at the University of Pittsburgh. The study took place at a large, tertiary care academic medical center with 175 specialty Graduate Medical Education programs. Our institution trains nearly 1800 residents and fellows with over 180,000 healthcare professionals participating in continuing education programs annually. The University of Pittsburgh School of Pharmacy has 2 patient simulators that are both wireless and mobile. Taking advantage of this mobility has allowed for integration of simulation training into residents’ onsite training in the hospital setting.

Twenty-four individual medical and pharmacy residents were recruited from a population of trainees at UPMC, Presbyterian campus. Investigators assigned recruited individuals to serve as part of a multidisciplinary pair (i.e., one physician and one pharmacist) based on schedule availability. Each multidisciplinary pair was assigned to either the cross-training or current training (control) condition, using a computer-based automated randomization scheme. Participants were blinded to which training condition they were assigned.

All participants provided informed consent and completed the entire study within one hour. All participants started by completing demographic questions and a six-question baseline knowledge assessment. Then, participants underwent the pre-training assessment scenario in their own professional role and completed a teamwork and interprofessionalism survey based on their experience in the pre-training scenario (Appendix A and Appendix B). Next, the participants completed the training portion of the study. The training portion consisted of a video-based content module, which trained them how to either intubate or perform pharmacokinetic calculation depending on randomization, and a simulation-based practice scenario, which allowed them to practice those skills in the collaborative treatment of an unstable critically ill simulated patient. During the training portion, clinicians randomized to cross-training were educated on an alternative professional’s role and allowed to practice that skill in simulation. Cross-trained physicians reviewed a video and then practiced pharmacokinetic dosing to provide an optimal phenytoin recommendation. Cross-trained pharmacists reviewed a video and then practiced intubating a patient requiring immediate airway protection. Participants randomized to the control group reviewed the video relevant to their own profession and remained within their own professional role in the training simulation scenario (i.e., the medical resident in the control group reviewed the intubation video and practiced intubation on the simulator). Then, participants underwent the post-training assessment scenario in their own professional role and completed a teamwork and interprofessionalism survey based on their experience in the post-training scenario. Finally, participants completed a final questionnaire with the same six knowledge assessment questions from the baseline assessment and one open-ended question about their overall impression of the experience. 

The primary outcome measures were teamwork and interprofessionalism assessed by change in self-reported teamwork and interprofessionalism scores from pre- to post- training. Teamwork questions were adapted from the survey validated by Heinemann and colleagues, which focused on individuals’ perception of the quality of care provided by a team and the quality of teamwork demonstrated [11,12]. Interprofessionalism questions were adapted from the scale validated by Luecht and colleagues, and included the following subscale areas: perceived need for cooperation, actual cooperation, and understanding others’ values [13]. 

A secondary outcome measure of content knowledge was assessed through a change in the number of knowledge questions answered correctly before and after participating in the study. These knowledge questions were written to investigate whether the overall study design also offered educational value to the residents.

Likert scale responses were coded to numerical values, with a response of strongly disagree corresponding to a 1 and strongly agree corresponding to 6. Across the 20-question survey, 9 questions corresponded to teamwork and 11 questions corresponded to interprofessionalism. Likert scale responses were aggregated to provide an overall score for teamwork and interprofessionalism for each participant. Maximum scores a participant could record for teamwork was 54 by recording a strongly agree (or 6) for each of the 9 relevant questions; similarly, maximum scores a participant could record for interprofessionalism was 66 by recording a strongly agree (or 6) for each of the 11 relevant questions. A Wilcoxon signed-rank test compared pre-team to post-team scores for each group (control and cross-training), respectively. A Wilcoxon signed-rank was also used to compare pre-interprofessionalism to post-interprofessionalism scores for each group. As a secondary endpoint, a brief content 6-question knowledge assessment was conducted. Pre-knowledge and post-knowledge scores were compared for each group using a paired T-test. The open-ended question underwent thematic analysis for trends in responses. 

## 3. Results

Demographic characteristics of participants are summarized in Table 1.

### 3.1. Teamwork and Interprofessionalism

Complete survey scores for teamwork and interprofessionalism are summarized in Table 2. Professional individuals that experienced cross-training showed a significant improvement in their average survey teamwork scores (42.8 (79.26%) v 46.1 (85.37%), *p* < 0.05), whereas individuals in the control group did not (44.1 (81.64%) v 45.8 (84.88%), *p* > 0.05). Cross-trained individuals’ average post-training team score increased by 6.11% compared to their pre-training score; for the control group, the average post-training score increased by 3.24% compared to their pre-training score. Neither group demonstrated a significant improvement in interprofessionalism scores (cross training +0.45%, control group −1.52%, *p* > 0.05); however, baseline interprofessionalism scores for both groups averaged at approximately 85%.

Complete survey scores for teamwork and interprofessionalism.

### 3.2. Knowledge Assessment and Thematic Analysis

All participants demonstrated a significant improvement in knowledge scores when comparing post-study knowledge assessment scores to baseline knowledge assessment scores (Table 3). Feedback from participants was extremely favorable with several comments addressing the value of simulation-based learning in the interprofessional setting. One participant stated “Great. Practical/realistic scenario with ideal availability of PharmD. Reminds me of how useful interdisciplinary work can be.” Other comments included, “Helpful. Gave me insight into my own deficiencies and reinforced my appreciation of the interdisciplinary approach to cases/medicine”, “Very interesting—made both of us think on our feet. My training helped choose correct therapies. I liked helping the MD choose doses for sedating agents and paralytics”, and “Overall it was a good experience to see some of the calculations involved with dosing certain agents. I tend to rely on the pharmacists and dosing algorithms in my day to day practice, so it was interesting to see the overall formulas.”

All participants demonstrated a significant improvement in knowledge scores when comparing post-study knowledge assessment scores to baseline knowledge assessment scores. Cross-trained individuals’ average knowledge scores increased by 14.00%; whereas, for the control group, the average post-knowledge score increased by 13.89%.

## 4. Discussion

Cross-training is an educational strategy that has been extensively studied and successfully incorporated into other disciplines, such as aviation and engineering, yet cross-training has not previously been used in the healthcare setting [1,9,10]. The objective of this study was to assess if the benefits of cross-training, primarily improved teamwork and interprofessionalism, could also be observed in healthcare professionals. Additionally, it was important to evaluate the content knowledge changes with the simulation education provided. 

We observed an improvement in teamwork amongst participants that experienced cross-training. Of note, we observed that participants that were cross-trained depended heavily on one another to complete the training scenario. For instance, pharmacists that were cross-trained to intubate needed physician assistance in order to achieve appropriate laryngoscope placement to allow for optimal visualization of the vocal cords. Likewise, physicians that were cross-trained on phenytoin kinetics still referenced the pharmacist to assure that they were selecting appropriate laboratory values to insert into the pharmacokinetic equation for dosing. Interestingly, we observed that both counterparts were then shifted into a teaching role in order to clearly explain their usual role, so that their cross-trained partner could complete their task to achieve stabilization of the simulated patient. 

While changes for interprofessionalism scores did not reach statistical significance for either group, scores trended up for the cross-training group, while scores trended down for the control group. In future studies incorporating cross-training, the impact on interprofessionalism should be further explored. Of note, we also selected from a population of participants that currently train together as residents, which may explain why a high baseline interprofessionalism score was observed in both groups. 

Overall, we saw an improvement in knowledge scores in both the control group and cross-training group. This suggests that the content of the simulation-based scenarios allowed for important concepts to be taught through simulation. Moreover, similar improvements in both groups suggest that the cross-training strategy did not interfere with the teaching of key concepts. This emphasizes the importance of review by expert panel to assure that the learning objectives for each simulation scenario are indeed achieved through the design of each simulation scenario. 

Since we chose to target residents as participants for our project, time was an important constraint to consider. We found that study investigators had to be flexible in terms of availability to set up the simulation to accommodate physician and pharmacist availability. Often, scheduled session times had to be adjusted or moved to alternative days based on patient care needs. We found that we were able to allow participants to reflect on their simulation sessions and discuss key points; however, we were not able to expand about ideas offered during debriefing (Appendix C). We believe that the flexibility of transporting the simulation equipment near patient care areas increased our participant recruitment by making our study more accessible; this decreased time invested by participants by not asking them to meet at a central location. Alternatively, recruiting current healthcare students may be an approach to consider as students often have more availability than residents.

Our study may have been influenced by selection bias, as participants that enjoy simulation-based activities would have been more likely to participate. However, we collected both pre- and post- interventional measures to limit the influence of selection bias on our study measures. 

Finally, our study was limited by a small sample size, since this was a pilot study. Notably, we did capture 92% of all pharmacy residents available, suggesting that we had optimal recruitment. Moreover, we did achieve statistically significant improvements in many of our outcomes. We recommend that cross-training be further studied within the healthcare setting to determine if these results can be replicated in a larger study population.

## 5. Conclusions

Our study demonstrates that cross-training may improve teamwork in interprofessional teams, when incorporated into simulation-based training sessions involving medical and pharmacy residents. Larger studies should evaluate this concept further to determine the value of cross-training within an interprofessionalism setting. 

## Figures and Tables

**Table 1 pharmacy-07-00013-t001:** Demographics.

Demographic	Cross Training Group, n = 12	Control Group, n = 12
Medicine Residency, n (%)	6 (50.0%)	6 (50.0%)
Pharmacy Residency, n (%)	6 (50.0%)	6 (50.0%)
Residency Completed, in years	2.1	1.7
Experienced Interprofessional Learning in Professional Degree Program, n (%)	10 (83.3%)	8 (66.7%)

Summary of demographic characteristics of medical and pharmacy resident participants.

**Table 2 pharmacy-07-00013-t002:** Summarized Survey Scores for Teamwork and Interprofessionalism; Reported as an Average Score, n (%).

	**Pre-Training Teamwork Score**	**Post-Training Teamwork Score**	***p*-Value**
Cross Training	42.8 (79.26)	46.1 (85.37)	<0.05 *
Control Group	44.1 (81.64)	45.8 (84.88)	>0.05
	**Pre-Training Inter-Professionalism Score**	**Post Training Inter-Professionalism Score**	***p*-Value**
Cross Training	56.5 (85.61)	56.8 (86.06)	>0.05
Control Group	55.7 (84.34)	54.7 (82.83)	>0.05

* Indicates a statistically significant outcome.

**Table 3 pharmacy-07-00013-t003:** Summarized Survey Scores for Knowledge; Reported as Average Score, n (%).

	Pre-Training Knowledge Score	Post-Training Knowledge Score	*p*-Value
Cross Training	4.7 (81)	5.7 (95)	<0.05 *
Control Group	4.8 (79.17)	5.4 (93.06)	<0.05 *

* Indicates a statistically significant outcome.

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
