# Peer review of "Effects of Cross-Training on Medical Teams’ Teamwork and Collaboration: Use of Simulation"

_pharmacy, 2019, doi:10.3390/pharmacy7010013_

Round 1

Reviewer 1 Report

Interesting study showing the importance of proper exchange of information and experience among different professions in a medical team. Everybody "profited" from the cross-training design.

Reviewer 2 Report

Overall very well written paper and interesting topic. 

few minor suggestions:

In methods, line 72 and 73 please reword that sentence to make it easier to read. 

line 87, there is an extra letter f in that line.

Line 101 there are some extra words near the end of the line.

How did you decide on the number of participants? This is discussed a little in the limitations section. 

Results:

for Table 1, did you run statistics on years in residency and IPE experience before this study? 

Discussion:

Were the increases you saw with teamwork similar to other professions? 

Interesting that knowledge scores increased significantly in the control group that basically did a review of their own skills. 

Appreciate seeing the scenarios and questions at the end of the article. 

Conclusion - I would recommend to reword this to fit the purpose of your study.

Reviewer 3 Report

Please see attached

Round 2

Reviewer 3 Report

Nice job addressing the reviewers feedback